# Electrically Tunable Hydrogen-Bonded Liquid Crystal Phase Control Device

**Ryota Ito \*, Michinori Honma and Toshiaki Nose**

Department of Electronics and Information Systems, Akita Prefectural University, Yurihonjo 015-0055, Japan; mhonma@akita-pu.ac.jp (M.H.); t_nose@akita-pu.ac.jp (T.N.)
**\*** Correspondence: r_ito@akita-pu.ac.jp; Tel.: +81-184-27-2249

**Abstract:** Terahertz waves have attracted much attention mainly because of their potential in imaging, security checking, nondestructive testing, and information and communication technologies. In the past few years, there has been an extensive effort to investigate terahertz wave control devices. Liquid crystal (LC) devices are strong candidates for high-performance terahertz wave control devices because of their controllability at low drive voltages and their low power consumption. In this study, we fabricated an electrically tunable phase control device by using a hydrogen-bonded LC material. We investigated the performance of the LC phase shifter by using a far infrared continuous wave laser. We also estimated the birefringence and absorption properties of the hydrogen-bonded LC at 2.5 THz by using Jones matrix calculations. The measurements and calculation results indicated that the hydrogen-bonded LC showed no dichroism at 2.5 THz. Based on the absorption properties, we believe that it could be a strong candidate for use in future terahertz devices.

**Keywords:** terahertz wave; liquid crystal; phase control device

## 1. Introduction

The terahertz region of the electromagnetic wave spectrum ranges from 0.1 to 10 THz. Over the last few decades, there has been considerable interest in terahertz applications, such as imaging, security checking, nondestructive testing, and in information and communication technologies [1]. In order to successfully achieve these terahertz applications, terahertz wave control devices, including a beam former, a lens, an antenna, and a polarization control device, are essential. It is beneficial if these devices are tunable and have low absorption loss. To realize tunability in terahertz wave control devices, variable-focus terahertz lens using pumping oil [2], electrostatically actuated diffraction gratings [3], magnetically tunable ferrofluid [4], electrically tunable metamaterial [5], electrically tunable graphene-based polarizer [6], and optically tunable terahertz filter [7] have been demonstrated.

Another advantageous possibility is the use of high-performance liquid crystal (LC)-based devices in a terahertz application due to its controllability at low drive voltage and its low power consumption without any mechanical movement. Nose and coworkers reported the optical properties of LCs in the terahertz frequency region using a far infrared (FIR) continuous wave (CW) laser in 1997 [8]. They revealed that LCs exhibited birefringence in the terahertz frequency region, and thus the LCs were shown to be a promising material for terahertz devices. Koch and coworkers reported the optical properties of LCs in a terahertz frequency region by using terahertz time-domain spectroscopy (TDS) [9]. They reported the properties of many LCs in the terahertz region [10–14] and in terahertz wave control devices [15–17]. Pan and coworkers also reported the terahertz properties of many LCs using terahertz TDS [18–23] and many types of terahertz phase control devices using the birefringence of LCs in the terahertz frequency region [24–28]. Several authors have reported the improvement of the LC-based terahertz phase shifter. A transparent electrode is very important for an electrically tunable

LC-based terahertz phase shifter. Pan and coworkers have reported tunable LC terahertz phase shifters by using indium-tin-oxide nanowhiskers as transparent electrodes [28]. Lu and coworkers also have reported a high transmittance tunable LC terahertz phase shifter using porous graphene electrodes [29]. Since the LC-based terahertz phase shifter needs a large cell gap (500 μm or more), improvement in response speed is an important subject. To achieve high speed operation, the introduction of a reflection structure [30], polymer-stabilized LC (PSLC) [17], and electrospun nanofiber [31] have been reported. In addition, the development of LC material is also important for high-performance terahertz LC phase shifters. Recently, LCs with high birefringence in the terahertz region have been reported by Lu [32] and Koch [14]. These two reports indicate the promise of high-birefringence LCs for use in LC-based terahertz devices. However, all the reported nematic LCs exhibited dichroism in the terahertz region. In other words, the absorption in the ordinary direction $\alpha_o$ was not equal to the extraordinary absorption $\alpha_e$. Previously, we investigated phase-shifting interferometry by using an LC phase shifter and reported that the dichroism of the LC caused unwanted variation in the intensity of the terahertz phase shifter [33]. Therefore, LC materials with no dichroism are significant not only in phase-shifting interferometry, but also in future terahertz applications.

In this study, we fabricated an electrically tunable phase control device by using a hydrogen-bonded LC material. We investigated the performance of the LC phase control device by using an FIR CW laser and estimated the birefringence and absorption properties of the hydrogen-bonded LC at 2.5 THz. We also simulated the operational properties of the LC phase control device by using Jones matrix calculations.

## 2. Materials and Methods

Figure 1 shows the molecular structure of the hydrogen-bonded LC. This material exhibited a thermotropic liquid crystalline phase due to the intermolecular hydrogen bonding between two identical molecules. Hydrogen bonding has an important role in supramolecular LCs [34]. In this study, a hydrogen-bonded liquid crystal (LC1) was introduced to a phase control device. LC1 is a mixture of molecules (as seen in Figure 1) that vary by alkyl chain length. We also studied a standard nematic LC (E44) for comparison.

**Figure 1.** Molecular structure of hydrogen-bonded liquid crystal.

Figure 2 shows the structure of an electrically tunable phase shifter for terahertz waves. The LC material was injected into a normal sandwich cell. Both inner surfaces of the substrates were treated with antiparallel rubbing after coating the planar alignment layer with polyimide (SE2170, Nissan Chemical Industries, Tokyo, Japan) to obtain homogeneous alignment. The cell thickness was determined by using sheet spacers. The thickness of the LC layer was 800 μm for terahertz operation. To keep the high transmittance of the terahertz wave, we used quartz substrates and Poly(3,4-ethylenedioxythiophene):Poly(styrenesulfonate) (PEDOT/PSS) electrodes. The direction of the LC molecules (director) could be controlled by an external electrical field.

Figure 3 shows the experiment setup. In this study, the terahertz wave intensity profiles were measured by using an FIR CW laser as a signal source. This laser was the major source of a coherent CW with a powerful terahertz radiation above 0.3 THz. Here a $CO_2$ laser was used for pumping the $CH_2F_2$ gas, and a frequency of 2.5 THz was used for our measurements. We placed the LC device between two wire-grid paralyzers, and the intensity of the terahertz wave was detected by a pyroelectric detector. To eliminate the influence of laser power variation, we checked the power of the THz wave by using

pyroelectric detector 1 at all times. An accurate transmittance was obtained by using the intensity of pyroelectric detector 2 normalized by the intensity of detector 1.

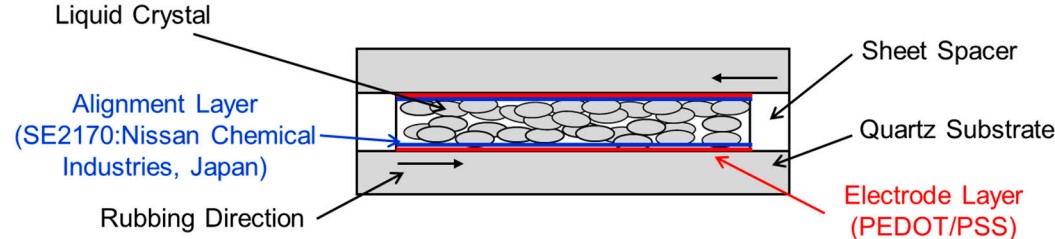

**Figure 2.** Structure of electrically tunable phase shifter for a terahertz wave.

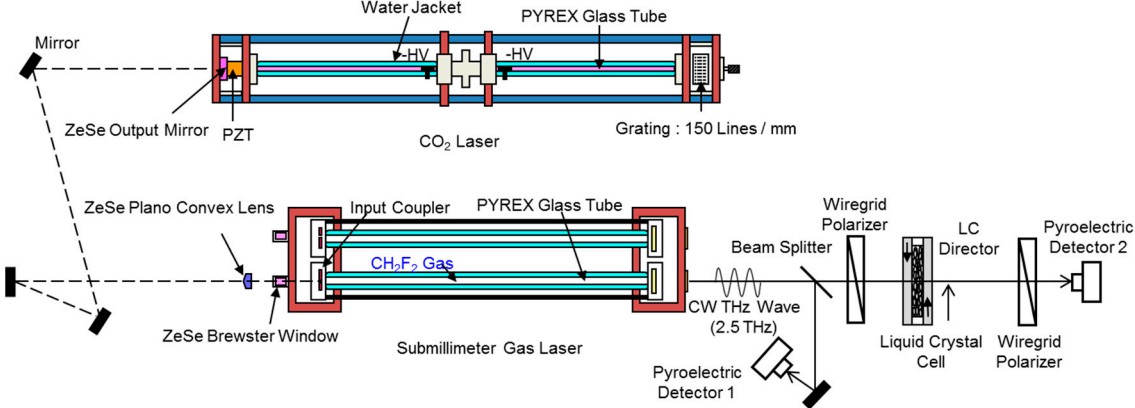

**Figure 3.** Experiment setup for terahertz measurements.

The polarization state of the transmitted terahertz wave through the LC device was calculated by using the Jones matrix method [35]. Figure 4 shows the Jones matrix calculation model of a homogeneous-alignment LC device in the terahertz region. In this situation, the electric fields of the transmitted terahertz wave $E_x$ and $E_y$ are described as follows:

$$\begin{bmatrix} E_x \\ E_y \end{bmatrix} = P_A W \begin{bmatrix} \cos \Psi_P \\ \sin \Psi_P \end{bmatrix}, \tag{1}$$

where $\Psi_p$ is the angle of the polarizer as defined in Figure 4. The Jones matrix of the analyzer $P_A$ and that of the homogeneous LC device $W$ in Equation (1) are written as

$$P_A = R(\Psi_A) \begin{bmatrix} 1 & 0 \\ 0 & 0 \end{bmatrix} R(-\Psi_A), \tag{2}$$

$$W = R(\Psi_i) \begin{bmatrix} exp\left(-\frac{i\Gamma}{2}\right) exp\left(-\frac{2\pi n_e'' d}{\lambda}\right) & 0 \\ 0 & exp\left(\frac{i\Gamma}{2}\right) exp\left(-\frac{2\pi n_o'' d}{\lambda}\right) \end{bmatrix} R(-\Psi_i), \tag{3}$$

where $\lambda$ is the wavelength of the incident terahertz wave, $d$ is the thickness of the LC layer, and $n''$ is the imaginary part of the complex refractive indices of the LC. Here, subscripts of $e$ or $o$ mean the extraordinary and ordinary refractive indices, respectively. The $R(\Psi)$ and $\Gamma$ are given as

$$R(\Psi) = \begin{bmatrix} \cos \Psi & -\sin \Psi \\ \sin \Psi & \cos \Psi \end{bmatrix}, \tag{4}$$

$$\Gamma = \frac{2\pi |n_e' - n_o'| d}{\lambda}, \tag{5}$$

where $n'$ is the real part of the complex of refractive indices of the LC.

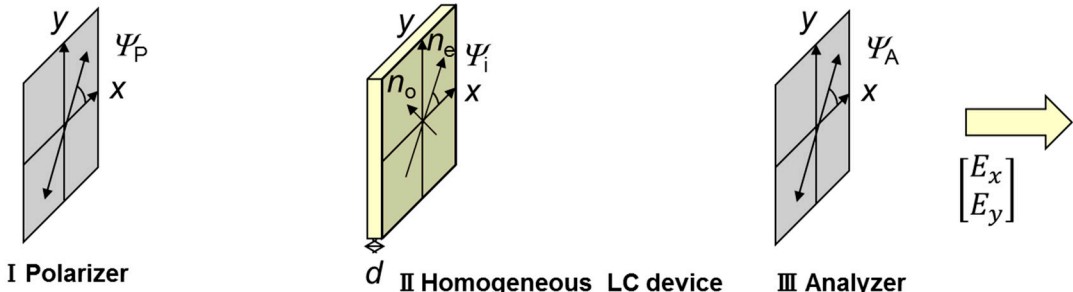

**Figure 4.** The Jones matrix calculation for the homogeneous alignment of the LC device in the terahertz region.

## 3. Results and Discussions

Figure 5 shows the transmitted terahertz intensity of LC1 as a function of the applied voltage. The data were obtained by using an FIR CW laser as shown in Figure 3. We placed the LC device between the polarizers of closed Nicols. The direction of the polarizers and the director of the LC device were $\Psi_P = 45°$, $\Psi_A = -45°$, and $\Psi_i = 0°$, respectively. The transmitted 2.5 THz wave was measured by varying the voltage applied to the LC device. Figure 5a shows the result from LC1, and the result of E44 is shown in Figure 5b for comparison.

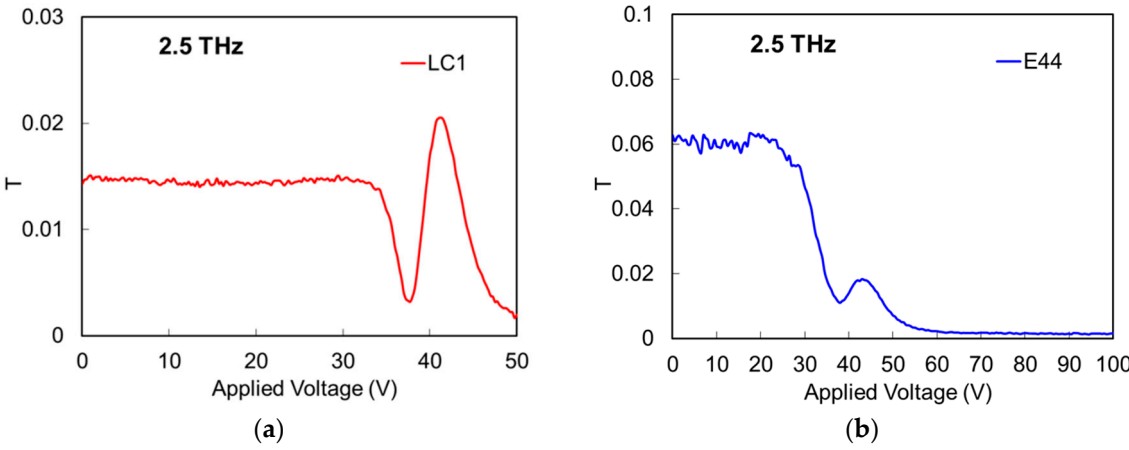

**Figure 5.** Transmitted terahertz intensity of (**a**) a hydrogen-bonded liquid crystal (LC1) and (**b**) a standard nematic LC (E44) as a function of the applied voltage.

We successfully observed the transmittance oscillation by increasing the applied voltage. The threshold voltage was high in both cases of LC1 and E44, which was due to the influence of the PEDOT/PSS electrodes. We believe that the threshold voltage will be lower once the fabrication process of the PEDOT/PSS electrodes has been optimized.

The oscillation of transmittance shown in Figure 5 arose from the variation in polarization after passing through the LC device. In the case of E44, the transmittance oscillated with decreasing intensity, as shown in Figure 5b. This is due to the difference in the absorption coefficient between the extraordinary and ordinary direction. In the case of E44, the extraordinary absorption coefficient was smaller than that of the ordinary directions [33]. The electric field in the $x$ direction felt the extraordinary absorption, and the electric field in the $y$ direction felt the ordinary direction of absorption at 0 V. However, upon increasing the voltage, the LC molecules rose up, and the electric fields in both the $x$ and $y$ directions gradually felt the ordinary direction of absorption. Since the ordinary absorption coefficient in E44 was larger than the extraordinary absorption coefficient, the total transmittance decreased. Conversely, the transmittance oscillation doses did not decrease in the case of LC1, as shown

in Figure 5a. These results indicate that LC1 had the same value of ordinary and extraordinary absorption coefficients.

We previously reported that the birefringence of E44 is 0.2 [33]. The thickness of the LC layer was 800 μm in this case, and the retardation $R$ of the E44 phase shifter was 160 μm. The frequency of the incident wave was 2.5 THz, which corresponded to a wavelength of 117.7 μm. This value means that two minima appeared in the transmittance versus applied voltage plot, which correlated with the result shown in Figure 5b. Since there were also two minima in Figure 5a, the birefringence of LC1 might have been about 0.2 at 2.5 THz. To estimate the values of the birefringent and absorption coefficients of LC1, we investigated the dependence of transmittance on the director angle $\Psi_i$.

Figure 6 shows the experimental and calculated terahertz transmittance of LC1 at 2.5 THz. Here, we placed the sample between the polarizers of a parallel Nicols, and then we rotated the sample. The graph shows the transmittance as a function of LC director angle $\Psi_i$. Since we set the direction of the polarizers $\Psi_P = \Psi_A = 0°$, the data for $\Psi_i = 0°$, $180°$, and $360°$ corresponded to the results of the electric field of the terahertz wave that was parallel to the LC director. On the other hand, the data when $\Psi_i = 90°$ and $270°$ showed the results of the terahertz wave that was perpendicular to the LC director. In Figure 6, the transmittances of these two cases were almost the same. This result indicates that the extraordinary and ordinal absorption coefficients of LC1 had the same value. The solid line in Figure 6 shows the calculated results of the terahertz transmittance by using the Jones matrix calculation. We set the parameters as follows: $\Psi_P = \Psi_A = 0°$, $d = 800$ μm. The calculation data showed a good fit with the experimental data when we set $\Delta n = 0.17$, $n_e'' = 0.022$, and $n_o'' = 0.060$. In this study, we took into account the attenuation of the two substrates by setting the absorption coefficient of the quartz substrates at $\alpha_s = 0.5$ cm$^{-1}$. This value was consistent with the reported value [36], and the transmittance decreased by 95% when the thickness of each substrate was 500 μm.

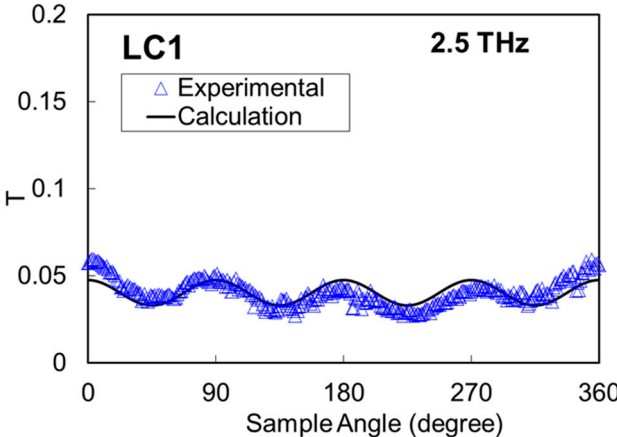

**Figure 6.** Experimental and calculation terahertz transmittance of LC1 at 2.5 THz.

Table 1 shows the birefringence, absorption coefficients, and dielectric anisotropy of LC1. In the case of LC1, which consisted of hydrogen bonding, $\alpha_e$ and $\alpha_o = 37$ cm$^{-1}$ and $\Delta n = 0.17$ at 2.5 THz. Here we also show the reported values of birefringence, absorption coefficients, and dielectric anisotropy for several LCs from References [8,12,22,32,33]. The birefringence of LC1 at 2.5 THz was not remarkably high compared to the LCs shown in Table 1. In addition, absorption coefficients of LC1 were slightly high or nearly the same as reported values of 2.5 THz in Table 1. It should be noted that the absorption coefficients $\alpha_e$ and $\alpha_o$ of the hydrogen-bonded LCs were almost of the same value. Since the hydrogen-bonded LC exhibits nondichroism, it is easy to design terahertz control devices the same way as with Si or other nondichroism materials [2–7]. On the other hand, the dielectric anisotropy of LC1 was 0.6, and this value was smaller than that of common LCs used for other terahertz applications. This result indicates that rise time (the switching time between off-state and on-state) of the LC phase shifter became slow compared to the cases of using the other LCs, as seen

in Table 1. We did not measure the relaxation time (the switching time between on-state and off-state), but we confirmed that the response of the LC1 phase shifter was almost same as with the case of E44. We believe that introduction of a reflection structure [30], a polymer-stabilized LC [17], and an electrospun nanofiber [31] is effective in improving the relaxation time of the LC1 phase shifter.

**Table 1.** Birefringence, absorption coefficients, and dielectric anisotropy of LC1. For comparison, reported values of various LCs are given.

| Liquid Crystal | $f$ (THz) | $\Delta n$ | $n_e''$ | $\alpha_e$ (cm$^{-1}$) | $n_o''$ | $\alpha_o$ (cm$^{-1}$) | $\Delta \varepsilon^6$ |
|---|---|---|---|---|---|---|---|
| LC1 | 2.5 | 0.17 | 0.035 | 37 | 0.035 | 37 | 0.6 (1 kHz) |
| E44 [1] | 2.5 | 0.2 | 0.022 | 23.5 | 0.06 | 64.1 | 16.8 (1 kHz) |
| K15 [2] | 2.5 | 0.165 | 0.0159 | 16.9 | 0.0379 | 40.4 | 20.07 (1 kHz) |
| E7 [3] | 1 | 0.14 | 0.015 | 3 | 0.035 | 7 | 13.8 (1 kHz) |
| NJU-LDn-4 [4] | 1.6 | 0.314 | 0.0526 | 12 | 0.0352 | 18 | 6.01 (1 kHz) |
| 1825 [5] | 2.5 | 0.371 | 0.0363 | 19 | 0.0262 | 13.7 | 17.0 (1.5 kHz) |

[1] Results from Reference [33]. [2] Results from Reference [8]. [3] Results from Reference [22]. [4] Results from Reference [32]. [5] Results from Reference [12]. [6] $\Delta \varepsilon = \varepsilon_{//} - \varepsilon_{\perp}$.

More detailed measurements are in progress to characterize the broadband terahertz optical properties of the hydrogen-bonded LC.

Figure 7 shows the simulation results of the polarization condition of the transmitted terahertz wave calculated by using the Jones matrix method. The transmitted terahertz wave polarization condition just after the LC phase shifter is displayed. We set the parameters as follows: $\Psi_p = 45°$, $\Psi_i = 0°$, $d = 800$ μm, and $\lambda = 117.7$ μm. Figure 7 shows the polarization conditions for retardation when $R = \Delta nd = \lambda$, $\lambda/2$, and $\lambda/4$, which correspond to phase shifts of 360°, 180°, and 90°, respectively.

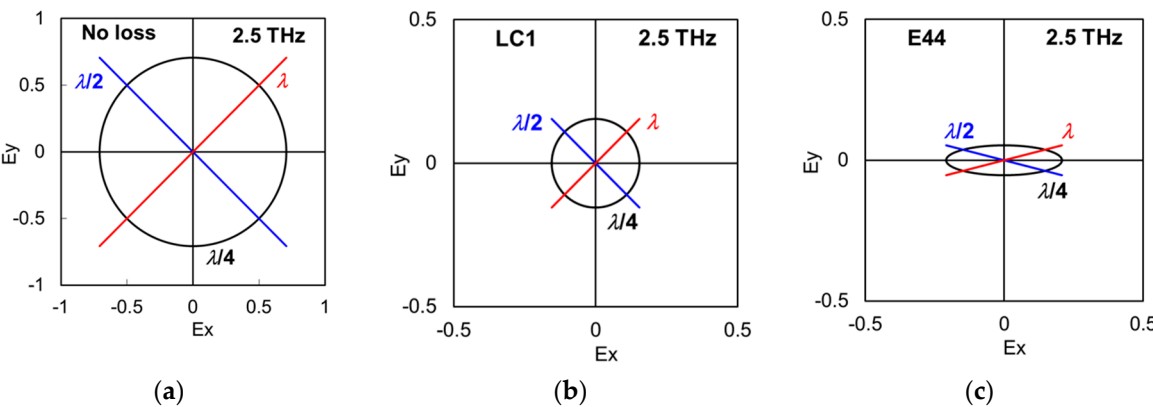

**Figure 7.** Simulation results of the polarization conditions of a transmitted terahertz wave. We set the absorption values for (**a**) zero (no loss), (**b**) LC1, and (**c**) E44. Retardation of the LC device was $\lambda$, $\lambda/2$, and $\lambda/4$.

Figure 7a shows the calculation results without losses for the ideal case. In this instance, the output terahertz wave was linearly polarized at $R = \lambda$ and $R = \lambda/2$ and was circularly polarized at $R = \lambda/4$. As shown in Figure 7a, the direction of the electric fields corresponded to the input terahertz wave at $R = \lambda$ and is 90° rotated from input terahertz wave at $R = \lambda/2$.

Figure 7b,c shows the calculation results by using LC losses, as shown in Table 1. In the case of LC1, the intensity of the output became lower, but a ±45° linearly polarized terahertz wave was obtained at $R = \lambda$ and $\lambda/2$. Conversely, since E44 exhibited dichroism at 2.5 THz, the direction of the output linear polarized terahertz wave deviated from ±45° at $R = \lambda$ and $\lambda/2$, as shown in Figure 7c. In addition, an exact circular polarized output was obtained at $R = \lambda/4$ in the case of LC1. However, the polarization of the output terahertz wave became ellipsoidal in the case of E44 because of its dichroism at 2.5 THz. These results indicate that a phase control device using LC1 was almost operating ideally at

2.5 THz because there was no dichroism. We believe that the hydrogen-bonded LC will be particularly useful in future LC-based terahertz devices.

## 4. Conclusions

In this study, we introduced hydrogen-bonded LC materials in an LC-based phase control device, and the transmittance properties were measured by using an FIR CW laser. The hydrogen-bonded LC phase control device was electrically tunable by using a PEDOT/PSS electrode. The birefringence and adsorption coefficients were estimated by fitting the experiment data using the Jones matrix calculation. We discovered that the extraordinary and ordinal absorption coefficients of the hydrogen-bonded LCs were almost of the same value. We also simulated the polarization conditions of the transmitted terahertz wave, which were calculated by using the Jones matrix method. In the case of the hydrogen-bonded LC device, the output terahertz wave polarization was $\pm 45°$ linearly polarized at $R = \lambda$ and $\lambda/2$, and circularly polarized at $R = \lambda/4$. On the basis of the absorption properties and the behavior of the hydrogen-bonded LC, we believe that it could be a strong candidate for use in future terahertz devices.

**Author Contributions:** R.I., M.H., and T.N. designed the experiments. R.I. performed the experiments and analyzed the data. This paper was mainly drafted by R.I. and checked and revised by M.H. and T.N.

**Funding:** This work was funded by the Nippon Sheet Glass Foundation for Materials Science and Engineering.

**Conflicts of Interest:** The authors declare no conflicts of interest.

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
