# Peer review of "Electrically Tunable Hydrogen-Bonded Liquid Crystal Phase Control Device"

_applsci, doi:10.3390/app8122478_

Reviewer 1 Report

Report on MS with reference #applsci-388960            

The paper entitled "Electrically tunable hydrogen-bonded liquid crystal phase control device" by Ryota Ito, Michinori Honma and Toshiaki Nose presents an experimental and theoretical study of a tunable phase control device based on liquid crystals (LCs) that is proposed for operation at THz frequencies.

The MS is globally well written and materials and methods well described. Nevertheless, I would recommend a check of the MS by an English eliminate some scarce grammar errors (for instance, “It is beneficial if these devices were tunable and have low absorption loss” in lines 29-30 of pages 1) and typos.

Results may be relevant to fabricate THz devices and, accordingly, I recommend the paper for publication. Nevertheless, a comparison to other structures/materials under research with a similar aim will be very useful for the reader.

To improve and place in context the results obtained in the paper authors must modify the MS according to the next two suggestions:

1 - Authors should introduce a short review of other structures/materials that are investigated to build THz phase shifters (such as metamaterials, graphene, etc.) and discuss how, at least potentially, the performance of the hydrogen-bonded LCs studied in the MS would compare to alternative technologies (both based in LCs and non-LCs).

2 - Authors should comment on the possible values of the response time of the devices investigated in the paper and compare it to the ones of other LC materials.

Author Response

We thank referees for careful reading our manuscript and for giving useful comments.

Our responses to the referee's comments are in Word file.

Reviewer 2 Report

Thank you very much for this very interesting paper on hydrogen-bonded LC and it's measurement at 2.5 THz. The realization of LCs with no dichoroism is a key feature for future LC-based components. Before publication I suggest the following modifications:

1) Please also add permittivity and tand values for the LC to enable a more easy comparison with other materials

2) Please add reference values for other LC materials, eg. commercial mixtures for comparison.

3) You stated, that the properties have been "estimated by fitting". Can you give some hints, which accuracy we can expect from your measurement setup.

4) Please add information on switching speed and viscosity of the material.

5) Can you estimate the phase shifter figure of merit based on your data. Please also add a plot where you show the phase-shifting behavior of your design.

Author Response

We thank referees for careful reading our manuscript and for giving useful comments.

Our responses to the referee's comments are in Word file.

Round  2

Reviewer 2 Report

Thank you very much for revising the paper.

The editing of the paper is good and the comments are addressed.

I personally thnik that you might consider revising your paper at a later instance in order to incorporate more recent results, e.g. "More detailed measurements are in progress to characterize the broadband terahertz optical properties" and switching times. The paper will benefit from these results.

Please check Fig.3 as it is truncated on the right side.

The results of your LC are promising but still not very good. Can you please comment on your ideas how to improve material properties. Especially when you consider recent published optimized LC, e.g . https://doi.org/10.1109/APUSNCURSINRSM.2017.8072651 .

"We did not measure the relaxation time (time of on sate to off state). But we confirm that response of LC1 phase shifter is almost same as the case of E44.": How can you confirm this, when you did not measure it?

Author Response

We thank referee for careful reading our manuscript and for giving useful comments.

I attach here our revised manuscript, as well as a point-by-point response to the reviewers’ comments.
